# A terahertz-driven non-equilibrium phase transition in a room temperature atomic vapour

C.G. Wade[1,2], M. Marcuzzi[3,4], E. Levi[3,4], J.M. Kondo[1], I. Lesanovsky[3,4], C.S. Adams [1] & K.J. Weatherill[1]

There are few demonstrated examples of phase transitions that may be driven directly by terahertz frequency electric fields, and those that are known require field strengths exceeding $1\,MV\,cm^{-1}$. Here we report a non-equilibrium phase transition driven by a weak ($\ll 1\,V\,cm^{-1}$), continuous-wave terahertz electric field. The system consists of room temperature caesium vapour under continuous optical excitation to a high-lying Rydberg state, which is resonantly coupled to a nearby level by the terahertz electric field. We use a simple model to understand the underlying physical behaviour, and we demonstrate two protocols to exploit the phase transition as a narrowband terahertz detector: the first with a fast ($20\,\mu s$) non-linear response to nano-Watts of incident radiation, and the second with a linearised response and effective noise equivalent power $\leq 1\,pW\,Hz^{-1/2}$. The work opens the door to a class of terahertz devices controlled with low-field intensities and operating in a room temperature environment.

[1] Joint Quantum Centre (JQC) Durham-Newcastle, Department of Physics, Durham University, Durham DH1 3LE, UK. [2] Clarendon Laboratory, University of Oxford, Parks Road, Oxford OX1 3PU, UK. [3] School of Physics and Astronomy, University of Nottingham, Nottingham NG7 2RD, UK. [4] Centre for the Mathematics and Theoretical Physics of Quantum Non-equilibrium Systems, University of Nottingham, Nottingham NG7 2RD, UK. Correspondence and requests for materials should be addressed to C.G.W. (email: christopher.wade@physics.ox.ac.uk)

Phase transitions consist of sharp changes in the macroscopic properties of a physical system occurring upon the smooth variation of an external driving parameter (for example temperature, electric/magnetic field, etc.). The technological applications of phase transitions are diverse, ranging from the storage of energy as latent heat[1], to the action of shape memory alloys[2]. However, phase transitions directly induced by terahertz frequency radiation are rare. Known examples either require electric fields of order 1 MV cm$^{-1}$ that can only be created transiently using pulsed sources[3,4], or rely on heating from the terahertz radiation to drive the phase transition indirectly[5]. Here, we report a non-equilibrium phase transition driven directly by a weak, continuous wave (CW) terahertz frequency field ($\ll$1 V cm$^{-1}$), six orders of magnitude smaller than required in other work[3,4].

The system described in this work consists of an atomic vapour under continuous optical excitation to an energy level with large principal quantum number $n$, a so-called Rydberg level. Phase transitions in such non-equilibrium Rydberg systems have been a subject of recent experimental[6–8] and theoretical[9–11] interest. The Rydberg phase transition induces optical bistability[6] and has been associated with the presence of an Ising-like critical point in parameter space[12], while the bistability can be interpreted as a hysteresis cycle across an underlying first-order line[13,14]. In these systems, the laser driving decouples the internal atomic degrees of freedom from the thermal fluctuations, ensuring that the electronic configuration of the atoms does not reflect the temperature of the vapour.

The abrupt changes in system properties in response to a weak terahertz field make our non-equilibrium system applicable as a sensitive room temperature terahertz detector[15]. Terahertz devices have seen rapid development in recent decades[16], with new technology based on media from super-conducting[17] and semi-conducting[18] solids to atomic vapour[19–21]. However, the traceable calibration of terahertz detectors, typically using cryogenic bolometers, still yields substantial uncertainties[22]. Measurements of atomic properties are readily reproducible, and therefore lend themselves naturally to measurement standards. Rydberg atoms were first used to read out mm-waves ($\simeq$300 GHz) three decades ago in beam-line experiments using field ionisation and subsequent ion detection as the Rydberg sensor[23]. Since then practical techniques have progressed significantly, most notably through the coherent optical detection of Rydberg states[24], allowing coherent control of Rydberg states in compact, room temperature vapour cells[25]. These developments have facilitated precision microwave and mm-wave electrometry[21,26–28], where Rydberg atom techniques promise to provide an SI traceable standard.

Here we use a weak, continuous wave (CW) terahertz-frequency field to drive the Rydberg phase transition, and show that the phase transition has the potential to enhance the sensitivity of Rydberg microwave detectors, extending the spectral range further into the terahertz band ($\simeq$650 GHz).

## Results

**Experimental setup**. Our experimental system is outlined in Fig. 1a. The atoms of a thermal caesium vapour are continuously driven to the $21P_{3/2}$ Rydberg energy level using a three-step ladder laser excitation scheme[29], consisting of probe, coupling and Rydberg lasers (Methods section). The vapour is monitored by photographing optical atomic fluorescence and by measuring the transmitted probe laser power, $p$, which increases when atoms are shelved in long-lived Rydberg levels or ionised[30]. We use the parameter $t = (p - p_0)/p_0$ to indicate the fractional change in transmitted laser power, where $p_0$ is the probe laser power transmitted when the Rydberg laser is far off resonance.

By controlling the frequency and intensity of the Rydberg laser, the vapour can be prepared in either of two distinct steady states. The first, which we refer to as 'Off', is characterised by the emission of weak green atomic fluorescence (Fig. 1b) and low probe laser transmission; the second, which we refer to as 'On', corresponds to increased probe laser transmission, and bright orange fluorescence (Fig. 1c). The bright orange fluorescence is due to optical decay from a large selection of Rydberg levels, indicating re-distribution of the atomic population[6,21]. As the Rydberg laser frequency is scanned we see hysteresis in the response, and the system undergoes abrupt changes as it switches from one state to another (Fig. 1d)[6,8]. Denoting positive (blue) laser detuning from the atomic line by $\Delta_R/2\pi$, the transitions from Off-to-On and On-to-Off occur at $\Delta_R = \Delta_+$ and $\Delta_R = \Delta_-$ respectively, making the system bistable for $\Delta_- < \Delta_R < \Delta_+$.

The response is strongly modified when a continuous-wave terahertz-frequency electric field is applied to the vapour. The field has a frequency of 0.634 THz and resonantly couples the $21P_{3/2}$ Rydberg state to the neighbouring $21S_{1/2}$ level. In particular, when the terahertz field is introduced the bistablity window shifts to a new range, $\Delta'_- < \Delta_R < \Delta'_+$, as shown in Fig. 1d. We define shift parameters, $\delta_- = \Delta'_- - \Delta_-$ and $\delta_+ = \Delta'_+ - \Delta_+$, and we note that in the example shown in Fig. 1d the hysteresis window shifts and narrows, $\delta_+ < \delta_- < 0$.

In order to demonstrate the terahertz field driving the phase transition directly, we set the laser frequency to the value denoted by the vertical dashed line in Fig. 1d, and hold all parameters constant while the terahertz electric field amplitude is varied. The vapour is initialised in the 'Off' state at zero terahertz field amplitude, and Fig. 1e shows the response as the terahertz field amplitude is slowly ramped up and then back down again. As the terahertz intensity increases we initially observe a quadratic rise in laser transmission (linear with intensity), before the system switches to the 'On' state, indicated by an abrupt increase in $t$. When the terahertz intensity is decreased again the vapour returns to the 'Off' state (accompanied by another sharp change in $t$), completing the full hysteresis cycle. The response of the system to sudden changes was observed to slow down close to the threshold intensity, in a manner which is qualitatively similar to that previously reported in Carr et al.[6]. The hysteresis cycle constitutes a strongly non-linear response of the system to the weak terahertz-frequency electric field.

To characterise the system we map the optical response to the terahertz field in the Rydberg laser- and terahertz-detuning $\{\Delta_R, \Delta_T\}$ plane for a selection of terahertz field amplitudes (Fig. 2). We describe the response by considering the regions in parameter space where the system is bistable. When the terahertz field is blocked (Fig. 2a), the bistability region is, as expected, independent of the terahertz field detuning. At intermediate terahertz field strength (0.15 V cm$^{-1}$, Rydberg transition Rabi frequency $\Omega_T^{meas}/2\pi = 31$ MHz) the bistable parameter space is deformed, and we see that $\delta_\pm < 0$ for $\Delta_T > \Delta_R$ and $\delta_\pm > 0$ for $\Delta_T < \Delta_R$ (Fig. 2b). When the field is 0.23 V cm$^{-1}$ $\Omega_T^{meas}/2\pi = 48$ MHz the bistable parameter space is split into two separate regions (Fig. 2c), and we note that bistability becomes absent for all $\Delta_T = \Delta_R$. The Rydberg transition Rabi frequency is recorded in a separate in-situ measurement of Autler–Townes splitting[26,31] (Methods).

**Mean-field model**. A full simulation of the microscopic dynamics involved in the phase transition is computationally unfeasible; however, previous work has been able to qualitatively describe the Rydberg phase transition using a mean-field model[6,7]. Here we use a similar model to capture the response of the system to the terahertz field (the full details of which are available in the

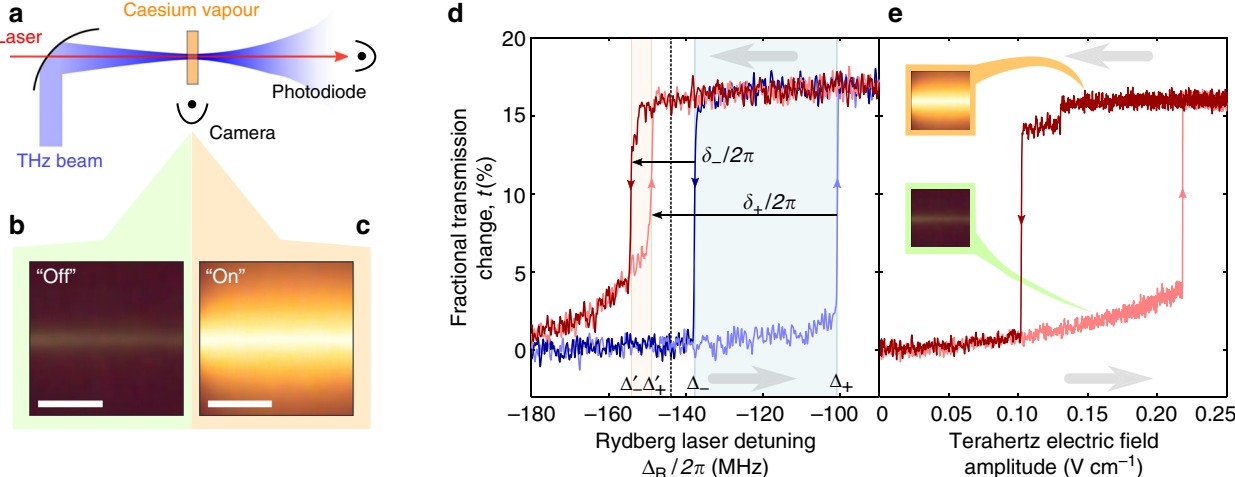

**Fig. 1** Experiment overview. **a** Experiment layout: A room temperature caesium vapour is continuously excited to a Rydberg energy level by laser driving, and manipulated by a co-axial THz field. The vapour is monitored by measuring laser transmission and atomic fluorescence. **b, c** Photographs of the atomic vapour when it is in the 'Off' **b** and 'On' **c** steady states. The scale bars are each 500 μm long. **d** Laser transmission with cycled laser detuning: The hysteretical system response (blue) is altered by the addition of a continuous wave terahertz field with amplitude 0.26 V cm⁻¹ (red). The frequency detuning range for which the response is bistable $\{\Delta_- \le \Delta_R \le \Delta_+\}$ shifts to $\{\Delta'_- \le \Delta_R \le \Delta'_+\}$, and the shift is parameterised by $\delta_\pm$ (defined in text). **e** Laser transmission with cycled terahertz power: The laser detuning is held at −145 MHz (vertical dashed line in **d**). The abrupt changes in laser transmission corresponds to the system switching between the 'Off' and 'On' phases (inset photographs). In **d** and **e** the grey arrows indicate the sense of change of the Rydberg laser detuning and the terahertz electric field amplitude, respectively

Supplementary Note 1). For simplicity, we start from the optical-Bloch equations for a single atom, and we label $|0\rangle$ the ground state, and $|R\rangle$ and $|T\rangle$ the two Rydberg energy levels, coupled by the THz field (Fig. 2d). As a first approximation we neglect the two intermediate states used in the experimental ladder excitation scheme, and consider a direct effective coupling between $|0\rangle$ and $|R\rangle$. The coherent part of the evolution is described (in a rotating-wave approximation) by the Hamiltonian

$$\hat{H} = \left[ \frac{\Omega_R}{2}\sigma_{0R} + \frac{\Omega_T}{2}\sigma_{RT} + h.c. \right] - D_R\sigma_{RR} - D_{RT}\sigma_{TT} \quad (1)$$

with $\Omega_{R(T)}$ the effective Rabi frequency of the laser (terahertz field), $D_{R(T)}$ the corresponding detuning, $D_{RT} = D_R - D_T$, and $\sigma_{ab} = |a\rangle\langle b|$ with $a, b \in \{0,R,T\}$. For the dissipative part, we phenomenologically describe depletion from the excited state $|R\rangle \rightarrow |0\rangle$ ($|T\rangle \rightarrow |0\rangle$) at a rate $\Gamma_R$ ($\Gamma_T$). In the following, we set $\Gamma_R = \Gamma_T \equiv \Gamma$, although the qualitative results of the simulation do not rely on this assumption. An observable $\mathcal{O}$ then evolves according to the Lindblad equation $\dot{\mathcal{O}} = i[\hat{H}, \mathcal{O}] + \sum_{\alpha=R,T} L_\alpha^\dagger \mathcal{O} L_\alpha - \{L_\alpha^\dagger L_\alpha, \mathcal{O}\}/2$, where the jump operator $L_\alpha = \sqrt{\Gamma_\alpha}\sigma_{0\alpha}$.

In recent experimental work evidence that the feedback mechanism responsible for the bistable behaviour derives from ionised Rydberg atoms was reported[8]. The study suggests that ions created by inter-atomic collisions generate electric fields within the vapour, which in turn alter the Rydberg excitation rate through Stark shifts of the atomic energy levels. To model the effect of ionisation we assume that a fixed fraction $q_{R(T)}/(1 + q_{R(T)})$ of the atoms in energy level $|R(T)\rangle$ spontaneously ionises, producing an ion density $n_{ions} = q_R \langle \sigma_{RR} \rangle + q_T \langle \sigma_{TT} \rangle$ of ions. We include mean-field shifts of the Rydberg levels $|R(T)\rangle$ in proportion to the ion density, which can be reabsorbed in the detunings via appropriate rescalings

$$\begin{aligned} D_R &\rightarrow D'_R = D_R - \alpha_R n_{ions}, \\ D_{RT} &\rightarrow D'_{RT} = D_{RT} - \alpha_T n_{ions}, \end{aligned} \quad (2)$$

where the coefficient $\alpha_{R(T)}$ is proportional to the polarisability of the $|R(T)\rangle$ energy level. We note that the 21S$_{1/2}$ state represented

by $|T\rangle$ is almost 20 times less polarisable than the 21P$_{3/2}$ state represented by $|R\rangle$ [32], and so we make the approximation $\alpha_T \approx 0$.

As an order parameter, we focus on the density of excited atoms $N = \langle \sigma_{RR} \rangle + \langle \sigma_{TT} \rangle$, which should provide an effective qualitative comparison to the experimental data, as the transmission $t$ monotonically increases with the number of atoms shelved in the Rydberg energy levels[30]. The procedure we use to estimate $N$ is described in the Supplementary Note 1 and consists of obtaining from the Lindblad equation the evolution of spin observables, replacing the detuning with the effective values in Eq. (2) and solving self-consistently for $\langle \sigma_{RR} \rangle$ and $\langle \sigma_{TT} \rangle$. The calculation results are shown in Fig. 2e–g, and we note that the model reproduces several important features of the experimental data: first, the bistability region appears at negative laser frequency detuning ($\Delta_R < 0$); second, the split in the bistability window occurs around the condition $\Delta_R = \Delta_T$ and; third, the upper bistability branch experiences a negative shift ($\delta_\pm < 0$), whereas the lower one a positive shift ($\delta_\pm > 0$). The match was obtained via a numerical scan of the parameters, and the plots in Fig. 2 correspond to $\alpha_R q_R = -8.3$ and $\alpha_R q_T = -5$ in units of $\Gamma$.

The underlying mechanism leading to Rydberg bistability has been a subject of debate. In cold atom ensembles the atomic energy level shifts that lead to Rydberg blockade (or anti-blockade in the opposite case)[33–36] are caused by dipole interactions, and this mechanism was initially invoked to explain the collective behaviour responsible for the vapour phase transition[6]. However, according to a recent work[37], pure van der Waals interactions among excited Rydberg atoms seem to be insufficient to support bistability in a randomly distributed gas, even when thermal atomic motion prevents the growth of fluctuation correlations. In the development of the mean-field model, we trialled terms in the equations arising from resonant dipole interactions. In the mean-field picture, these pair-wise interactions are incorporated by rescaling

$$\begin{aligned} D'_R &\rightarrow D''_R = D'_R - \varepsilon n_R, \\ D'_{RT} &\rightarrow D''_{RT} = D'_{RT} - \gamma n_T, \end{aligned} \quad (3)$$

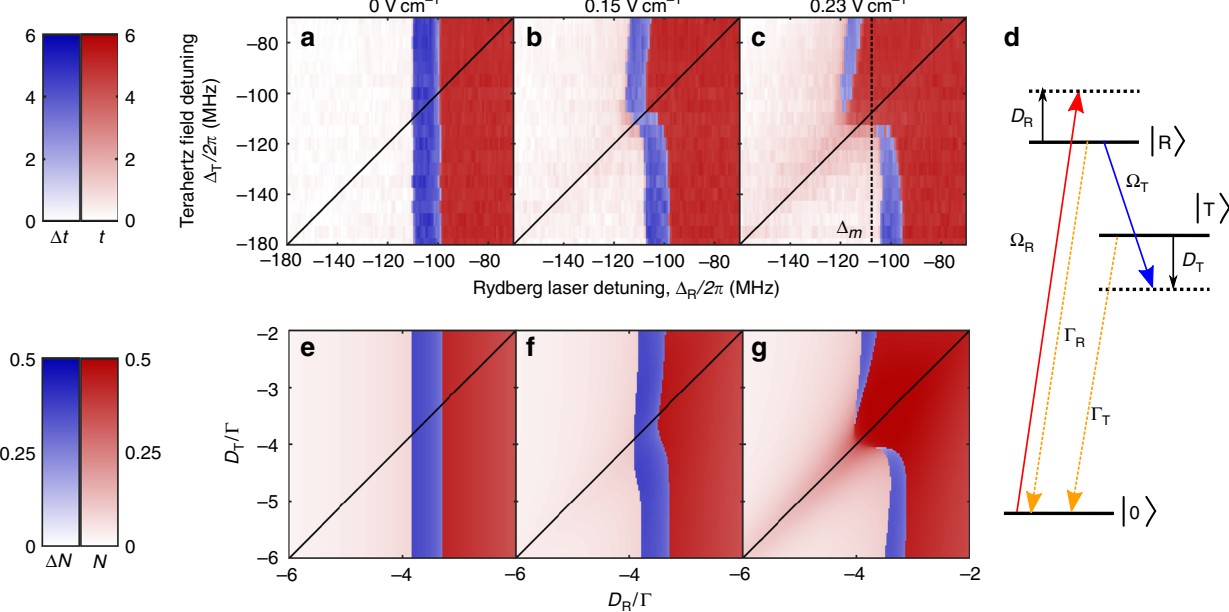

**Fig. 2** Experiment and theory comparison. **a–c** Experimental phase maps in laser/terahertz detuning: In areas where the system is monostable (red), we show the fractional increase in laser transmission, $t$ (%). Where the system is bistable (blue) we show $\Delta t = t_{On} - t_{Off}$, where $t_{On}$ ($t_{Off}$) is the fractional transmission change for the 'On' ('Off') state. In **a**, **b** and **c** the terahertz electric field has amplitude {0, 0.15, 0.23} V cm$^{-1}$, respectively, corresponding to Rydberg transition Rabi frequency $\Omega_T^{meas}/2\pi = \{0, 31, 48\}$ MHz. The solid black lines show the condition $\Delta_T = \Delta_R$, and the dashed black line in **c** shows the frequency at which the laser could be stabilised in order to demonstrate a reversible latch, $\Delta_R = \Delta_m$, (see the Sensing applications section). **d** Theoretical Model: Atoms are excited from $|0\rangle$ to $|R\rangle$ via a laser of Rabi frequency $\Omega_R$ and detuning $D_R$. A second transition takes them from $|R\rangle$ to $|T\rangle$ by means of a terahertz field of corresponding parameters $\Omega_T$ and $D_T$. Excited atoms spontaneously decay to $|0\rangle$ from the two levels $|R\rangle$ and $|T\rangle$ with rates $\Gamma_R = \Gamma_T = 1$. **e–g** Numerically calculated phase maps in laser/terahertz detuning: In areas where the system is monostable (red), we show the sum $N = \langle \sigma_{RR} \rangle + \langle \sigma_{TT} \rangle$ of the excited energy level populations. Where the system is bistable (blue) we show $\Delta N = N_{On} - N_{Off}$. The simulation parameters are fixed as follows: $\Omega_R = 1$, $\alpha = -8.3$, $\beta = -5$, $\gamma = \varepsilon = 0$, while $\Omega_T$ takes the three values 0 (**e**), 0.3 (**f**) and 0.8 (**g**). All parameters are given in units of $\Gamma_R = \Gamma_T \equiv \Gamma$ and are defined in the text

where $\varepsilon$, $\gamma$ are phenomenological parameters characterising the strength of the interactions. However, when these terms are dominant ($\alpha_{R(T)} \ll \varepsilon$, $\gamma$) the simulation does not match the behaviour observed in the experiment. Specifically, both branches display a positive shift $\delta_{\pm} > 0$ and, furthermore, if $\gamma$ is very large then the break in the bistability window does not occur at $\Delta_T = \Delta_R$. This suggests that dipole interactions do not dominate and instead ionisation plays the leading role in the feedback responsible for the phase transition, in agreement with previous work[8]. We note that the ionisation rate is dependent on vapour pressure and hence the Rydberg phase transition is strongly influenced by the temperature of the vapour.

**Sensing applications**. Room temperature atomic vapour holds particular promise as a medium for measuring terahertz fields. The well-known atomic properties allow for absolute calibration to SI units[26], and rapid recent progress in Rydberg electrometry[28,38,39] has delivered unprecedented sensitivity in the microwave[26] and mm-waves range[40]. This technique has been shown to be only limited by the shot noise of the probe laser[39], the intensity of which needs to be kept low to avoid spectral-broadening effects. Here, we report proof-of-principle terahertz field measurements that harness the collective behaviour of the vapour. Near the phase transition the abrupt changes in the optical response of the medium provide sharp spectral features even at high probe laser intensity, yielding a fast and sensitive method for measuring narrowband terahertz radiation.

When configuring the system as a detector, the complexity of the phase diagram provides the opportunity to exploit unconventional measurement protocols. Cycling the terahertz intensity can

result in a complete hysteresis loop, however, this is not necessarily the case. If the laser frequency is set so that the system is bistable when the terahertz field has zero intensity ($\Delta_- < \Delta_R < \Delta_+$), the hysteresis loop opens and we see a latching response (Fig. 3a). In this case $t$ increases steadily as the terahertz intensity is ramped up, until the system undergoes the transition to the 'On' phase, giving a sharp increase in $t$. However, if the terahertz field intensity subsequently returns to zero, the transition back to the 'Off' state is absent. Instead the system remains in the 'On' state for as long as the control parameters do not drift, effectively latching in an altered state. To gain full control of the 'On' and 'Off' states of the vapour using the terahertz field alone, it would be necessary to stabilise Rydberg laser to a detuning, $\Delta_m$, that divides the two branches of the bistable parameter space when terahertz field is at maximum intensity (indicated by the dashed line in Fig. 2c). In this circumstance we would expect a pulse of terahertz radiation with $\Delta_T > \Delta_m$ to transfer the system from 'Off' to 'On' (as we demonstrate with the latching detector configuration), and a pulse with $\Delta_T < \Delta_m$ to reverse the operation, taking 'On' back to 'Off'.

The result of implementing the system as a latching detector is shown in Fig. 3b. After the vapour is initialised in the 'Off' state, a 1 ms, 0.9 Wm$^{-2}$ terahertz pulse is 'detected' by the vapour, which switches to the 'On' state. The detector is then reset some time later by switching the laser power off for 1 ms. We present measurements of the laser transmission, but the phase transition is also very clear to the eye of an observer, as the vapour fluorescence changes from pale green to bright orange (Fig. 1b, c). The 1 ms gate time of the terahertz pulse was as short as we could achieve with our equipment, however, this does not reflect the

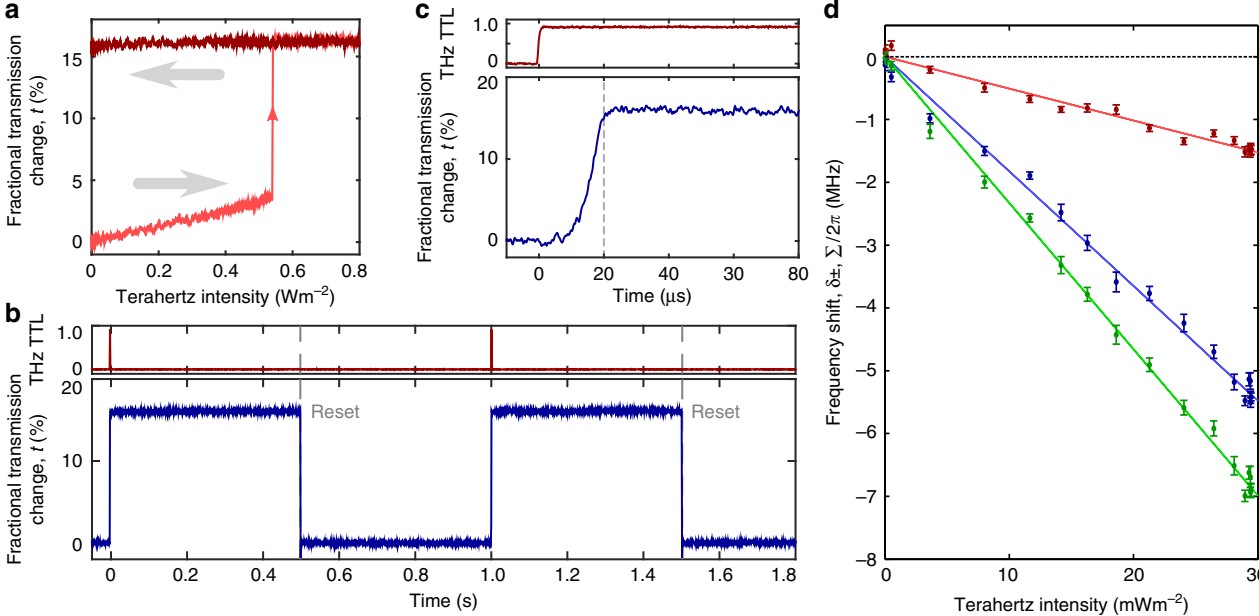

**Fig. 3** Sensing configurations. **a** Latching configuration: We show the laser transmission as the terahertz power is cycled (indicated by the grey arrows). Once a critical terahertz intensity has been exceeded the system latches in the 'On' state characterised by increased laser transmission. **b** Latching detector protocol: Having initialised the system in the 'Off' state, a 1 ms, 0.9 Wm$^{-2}$ terahertz pulse is 'detected' and flips the system from 'Off' to 'On'. The system remains in its altered state until the system is reset by cycling the laser power. **c** Latching response time: We show the same latching response on a microsecond timescale. Although the terahertz pulse is constrained to last 1 ms we see that the vapour takes only 20 μs to respond. **d** Frequency shift of bistability boundaries: The frequency shifts $\delta_+$ (blue), $\delta_-$ (red) and $\Sigma = \delta_+ + \delta_-$ (green) each show a linear dependence on the THz intensity. The error bars show the statistical uncertainty (standard deviation) in the mean of 10 or 11 repeated measurements, each lasting 1 ms

response time of the vapour. When the terahertz field is introduced, the vapour takes only 20 μs to switch to the 'On' state (Fig. 3c), indicating that a short period of weak illumination by terahertz radiation can permanently alter the collective state of the system. Taking the sensitive area as the probe laser beam cross section (1/e$^2$ radius 60 μm), we calculate that a minimum terahertz pulse energy ≈200 fJ is required to latch the vapour. A linear detector attempting to time the arrival of a single 200 fJ, 20 μs pulse would require a minimum bandwidth of 50 kHz and a maximum noise equivalent power (NEP) of 50 pWHz$^{-1/2}$. These requirements can be met comfortably by cryogenically cooled terahertz detectors[41], but stretch the ability of even high-performance room temperature devices[42–45]. Further work stabilising the Rydberg laser frequency to an atomic reference would allow the vapour to be biased closer to the phase transition, and therefore require a lower terahertz field intensity to latch the system.

Finally we show how to implement a detector with linear response, which—beyond speed and high sensitivity—is often a desirable property. To linearise the detector output we demonstrate a separate protocol, making use of the frequency shift of the laser detuning range for which the vapour is bistable. The terahertz field intensity is measured in situ by reading out Autler–Townes splitting induced by the terahertz field[26,31], and we repeatedly scan the Rydberg laser detuning (1 ms per cycle) and read $\Delta'_\pm$ from each scan. The shifts $\delta_+$, $\delta_-$ and $\Sigma = \delta_+ + \delta_-$ are then calculated using a reference measurement of $\Delta_\pm$. With the terahertz detuning set to $\Delta_T/2\pi = -91$ MHz such that $\Delta_T > \Delta_-$, we show the dependence of the shifts on the terahertz intensity in Fig. 3d (error bars show the standard error in the mean of repeated measurements). The frequency shifts follow a linear relationship with intensity, and by fitting straight lines constrained to pass through the origin[46] we deduce slope coefficients, $m_{\delta_\pm,\Sigma}$, which we combine with the average error of the data points $\bar{\sigma}$ and the measurement time, $\tau$, to

find an effective intensity NEP, $\sqrt{\tau}\bar{\sigma}/m_\Sigma = 48 \pm 3\ \mu$Wm$^{-2}$Hz$^{-1/2}$. Taking the detector area as the probe laser beam cross section yields NEP ≤1 pWHz$^{1/2}$, though it is not clear how the noise will scale with the laser beam area. Consecutive measurements at 2 ms intervals show no correlation, indicating that the noise present in the system is white in character. We note that the sensitivity of the detector to electric fields (rather than intensity) is not suitable to characterise the system, because the response is non-linear in the electric field strength $\left(E = \sqrt{2I/\epsilon_0 c}\right)$. In comparison to a detector that has a linear response to the electric field, the sensitivity of our system is suppressed for small fields but enhanced for strong fields.

In conclusion, we have demonstrated a phase transition in a thermal atomic vapour driven by a weak (≪1 Vm$^{-1}$) terahertz-frequency electric field. The necessary field strength is smaller than reported in other systems by over 6 orders of magnitude[3,4]. The strong, non-linear response is due to both the inherent inter-particle interactions in the vapour, and the large electric dipole coupling between the Rydberg atoms and the terahertz-frequency field. Non-linear effects induced by terahertz fields have been extensively studied[47], with applications ranging from non-linear spectroscopy[48] and high-harmonic generation[49], to the search for ferroelectric domain switching[50]. Yet such demonstrations rely on high intensity pulsed terahertz sources. By working in the vicinity of a phase transition we have shown a non-linear response to terahertz radiation in the CW regime, including permanent alteration of the state of the system.

The system can be configured as a narrowband terahertz detector, already showing performance comparable to state-of-the-art room temperature terahertz detectors[41]. While the system would not be suitable for broad-band terahertz sensing, transitions between different Rydberg states span the terahertz-frequency spectrum[21], and a detector could be 'tuned' to any of these frequencies by changing the Rydberg state to which the

**Table 1 Experimental parameters**

|  | Unit | Figs. 1, 3a–c | Fig. 2 | Fig. 3d |
|---|---|---|---|---|
| Vapour temperature | °C | 71 | 77 | 71 |
| Probe laser $1/e^2$ radius | mm | 0.06 | 0.03 | 0.03 |
| Coupling laser $1/e^2$ radius | mm | 0.05 | 0.10 | 0.10 |
| Rydberg laser $1/e^2$ radius | mm | 0.06 | 0.13 | 0.13 |
| Probe laser power | μW | 40 | 70 | 30 |
| Coupling laser power | μW | 60 | 30 | 140 |
| Rydberg laser power | mW | 330 | 310 | 410 |

atoms are driven. In general higher energy Rydberg states would require more laser excitation power and correspond to lower terahertz frequencies, but result in stronger interactions (faster ionisation and stronger polarizability). We anticipate further applications combining the phase transition with Rydberg electrometry[26] and Rydberg-fluorescence terahertz imaging[21].

## Methods

**Atomic vapour**. The caesium is contained in a quartz cell, with laser path length of 2 mm. The temperature of the vapour is stabilised around 70 °C using a servo circuit controlling either of two ovens which encase the glass cell, one constructed from stainless steel, the other from Teflon. The vapour temperature is inferred by measuring the transmission spectrum of the probe laser[51].

**Laser excitation**. We use a three-step excitation process to excite caesium atoms to the Rydberg state. The probe laser (852 nm) excites atoms to the $6P_{3/2}$ state, and the coupling laser (1470 nm) takes the atoms from the $6P_{3/2}$ state to the $7S_{1/2}$. Both the probe and coupling lasers are stabilised to the atomic resonances using polarisation spectroscopy[52]. The Rydberg laser (799 nm) is tuned to the $7S_{1/2}$ to $21P_{3/2}$ state transition, and is stabilised to a reference etalon. All three laser beams are co-axial and the Rydberg laser propagates in the opposite sense to the probe and coupling beams, minimising the 3-photon Doppler shift due to atomic motion through the optical fields.

**Terahertz beam**. The terahertz beam (0.634 THz) is generated from a microwave signal using an amplifier multiplier chain (AMC), manufactured by Virginia Diodes Inc. The beam is linear polarised to match the polarisation of the Rydberg laser and couples the $21P_{3/2}$ state to the $21S_{1/2}$ state. The terahertz beam propagates along the axis of the laser beams. We reference the terahertz field amplitude by making a direct, in-situ measurement of the Rabi-driving frequency of the Rydberg transition driven by the terahertz field. Combining the Rabi frequency with knowledge of the atomic dipole matrix element[32] then allows a calculation of the field amplitude. The measurement is performed by setting the terahertz field detuning to zero ($\Delta_T = 0$) and reading out the frequency interval between a pair of spectral features in the probe laser transmission (Autler–Townes splitting[31]). The result is an absolute measurement of the electric field amplitude, which can be traced directly to fundamental units.

**Automated control**. The main experimental parameters (power and detuning of the Rydberg laser and terahertz beams) are controlled from a computer using a LabView program. The microwave source and terahertz AMC are controlled directly through their respective interfaces, and the Rydberg laser frequency is controlled in two ways: For slow frequency scans (Figs. 1, 2 and 3a) the computer scans the reference etalon to which the Rydberg laser frequency is stabilised. For fast frequency scans (Fig. 3d) an acousto-optic modulator (AOM) is used instead, however, the range is limited to ≤100 MHz. The power of the Rydberg laser is controlled using the same AOM. The automated control allowed fast data collection, permitting the data shown in Fig. 2 to be recorded in only a few minutes. The data were recorded on separate occasions, with parameters summarised as shown in Table 1.

## Data availability

All data are available at the Durham University Collections database at https://doi.org/10.15128/r2s1784k74g.

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

## Acknowledgements

The research leading to these results has received funding from: EPSRC (Grants EP/M014398/1, EP/M013103/1 and EP/M013243/1, 'Networked Quantum Information Technology' Hub, NQIT); the European Research Council under the European Union's Seventh Framework Programme (FP/2007-2013) / ERC Grant Agreement No. 335266 (ESCQUMA); FET-PROACT project 'RySQ' (H2020-FETPROACT-2014-640378-RYSQ); Durham University; and The Federal Brazilian Agency of Research (CNPq). I.L. gratefully acknowledges funding through the Royal Society Wolfson Research Merit Award. We thank Mike Tarbutt, Andrew Gallant and Claudio Balocco for the loan of equipment, and Nikola Šibalić for stimulating discussions.

## Author contributions

The investigation was conceived by K.J.W. and C.S.A. The experiments were designed by C.G.W. and J.M.K. and performed by C.G.W. The simulation was performed by M.M., E.L. and I.L. All the authors contributed to writing the manuscript.

## Additional information

**Competing interests:** The authors declare no competing interests.

