## [Peer Review File · Nature Communications]

Reviewers' comments:

Reviewer #1 (Remarks to the Author):

The authors report on a new kind of terahertz-light-induced phase transition in an atomic vapor. The paper consists of two parts. In the first part, the author reports on detailed experimental results and their theoretical interpretation concerning the phase transition. In the second part, the authors mention about the new mechanism can be used as a sensitive terahertz detector.

First of all, I found their experimental work very interesting, while I would like to know the difference between this work and their previous work.

(1)The authors reported in Ref. [6] (in Fig. 4) that they observed the bistability and hysteresis window of the transmission by changing the Rydberg laser intensity. What is the crucial difference of the physical mechanism between their previous work in Ref. [6] and this work?

(2)Do they also observe the critical slowing down phenomena in the terahertz experiment; i.e., when they slightly change the terahertz intensity and observe the recovery, dose the recovery time diverge? This experiment may become the final proof that they surely succeeded to observe the terahertz - induced phase transition.

After the above questions are clarified, the paper may be suitable to publish in Nature Communications.

Reviewer #2 (Remarks to the Author):

The authors discuss their investigation of optical bistability in a room temperature cell of Rb atoms. As described in a previous publication [Ref 6] by the same principal investigators, when atoms in the cell are laser excited to the 21P Rydberg state, both the optical transmission and fluorescence from the cell exhibit sharp transitions between low and high values as a function of laser detuning from the Rydberg resonance such that the system exhibits bistability in these characteristic observables vs detuning. The sharp transitions between low and high transmission/fluorescence are attributed to a phase-transition. The addition of a colinear narrow-band THz beam introduces a coupling between the 21P and 21S Rydberg states, thereby modifying the range of detunings at which bistability occurs. Because the degree of modification depends sensitively on the THz field amplitude (and THz frequency), the authors propose to use the effect as a THz detector. Using a mean-field model, the authors attribute the bistable response to the optical and THz field combination as the result of microscopic electric fields, produced by collisional ionization of highly-excited atoms in the vapor, which shift the energies of the Rydberg levels coupled by the optical and THz fields. I find both the previously identified bi-stability, and its modification by the THz coupling reported here, to be very interesting effects and the experimental observations are striking. The proposal to use the effect as a THz sensor is, to my knowledge, novel and potentially important.

While the manuscript is generally well written, and I do not doubt the correctness of the measurements or their interpretation, I do have a number of comments and questions about the work that I believe are relevant to its ultimate impact in fundamental science and detector development.

1. The title, as well as much of the abstract and introduction, focuses on the THz-driven phase-transition theme, although this notion itself is not critical for the THz-sensor applications that seem to be the primary interest of the authors. Sharp transitions and bistability are important, regardless of

their source/interpretation. Further, I disagree with the claims in the abstract and introduction: "Here we report a room-temperature phase transition..." As the authors admit later in the introduction, the electronically excited system is non-thermal and, therefore, cannot be characterized by a temperature. The atoms may be at room temperature, but the bound electrons responding to the THz field are certainly not. Comparing the work to studies of phase-transitions (THz driven or otherwise) in thermal systems where the ratio of the interaction energy to the temperature is an important figure of merit, does not seem appropriate.

2. The authors focus on two specific Rydberg states. Have analogous phenomena been observed for other states? If so, why/how were the states which were used chosen? If not, why not? At least a brief discussion of these points would be instructive.

3. The observed effects depend critically on the THz frequency, thus it seems that a THz sensor based on this system would also only operate for a very well-defined THz frequency. Depending on the answer to 2 above, I suspect it would be possible to develop sensors for use at specific, narrow-band discrete frequencies, but not for a continuous band of frequencies or for broader band THz sources. Some discussion of the spectral range and sensitivity of a such a detector should be included.

4. Given that the bistability is attributed to electric fields due to collisional ionization, some discussion of the influence of external electric fields, including macroscopic fields due to longer lived space charge in the cell, should be included. On a related point, how is the collisional ionization (and therefore the bistability parameters and detector characteristics) affected by other variables such as: (i) the use of different Rydberg states as the system, or (ii) Rb temperature/pressure variations? The authors note that better experimental stability would improve the detector characteristics. Which of the numerous experimental parameters is the biggest limitation on the performance of the two different sensor schemes described: laser frequency/power stability, THz frequency stability, cell temperature, stray electric fields,....?

5. In the mean-field model, why are the spontaneous decay rates Γ_R and Γ_T chosen to be the same? The spontaneous decay rates for Rb S and P Rydberg states are well known and they are quite different from each other. How are the results of the model different if these decay rates are not identical?

6. In the mean-field model, how sensitive are the general results to changes in the collisional ionization rates? The authors state that sensors based on the bistability phenomenon will not be limited by laser power fluctuations as other techniques are. However, the collisional ionization rates will certainly vary with Rydberg density, which will vary with excitation laser and THz power variations. Has this effect been considered and analyzed?

Reviewer #3 (Remarks to the Author):

In this manuscript, the authors studied the phase transition of room-temperature atomic cesium vapor switched by weak THz-frequency electromagnetic pulse. They implemented Rydberg atoms in 21P energy level through three-photon process and then coupled to 21S energy level via 0.634 THz field. In this case, the cesium vapor showed a bistability of two steady states (low and high densities in Rydberg energy levels) and the two steady states behaved differently in laser transmission and also in fluorescence, upon being transmitted with the probe laser pulse (one of the three-step excitation lasers to 21P). The authors studied this bistability as a function of detunings of THz field and optical field. The result shows sensitive detection of THz field in V/cm-level of E-field amplitude, sensitive enough for a narrowband terahertz-field detector.

The measurements were compared with theoretical calculation using mean-field model taking into account the recently suggested inter-atomic collisions due to ionized Rydberg atoms [Ref. 8], showing an excellent replica qualitatively well agreeing the data. The quantum many-body dynamics is an intriguing question in physics which is being received much recent attention, especially in cold atom experiment. The present manuscript deals with the room-temperature bistability behavior of the Rydberg matter and opens the door to application of this fundamental physics to a real sensing application. The present work is in my opinion valuable and timely, and likely to be of interest for a broad audience.

Overall the manuscript is well written and understandable, and the method is accessible even for non-specialists. For all these reasons, I recommend publication in Nature Communications.

I only have a few minor comments that could help improve the manuscript a bit.

1. Most likely the result is not sensitive to the temperature, but it is worthy at least to mention it in the manuscript.
2. Probably the Ω^{meas}_T in Page 3 is the "measured" Rabi frequency between $|R\rangle$ and $|T\rangle$, but this is not defined and needs to explain how to measure it.

Reviewer #1:

The authors report on a new kind of terahertz-light-induced phase transition in an atomic vapor. The paper consists of two parts. In the first part, the author reports on detailed experimental results and their theoretical interpretation concerning the phase transition. In the second part, the authors mention about the new mechanism can be used as a sensitive terahertz detector.

This is an accurate summary of the work.

First of all, I found their experimental work very interesting, while I would like to know the difference between this work and their previous work.

(1) The authors reported in Ref. [6] (in Fig. 4) that they observed the bistability and hysteresis window of the transmission by changing the Rydberg laser intensity. What is the crucial difference of the physical mechanism between their previous work in Ref. [6] and this work?

The ionisation mechanism responsible for the collective behaviour is the same as that which leads to the previously observed Rydberg optical bistability (Ref. [6]), but here we instigate the phase transition with a terahertz frequency electric field (rather than by changing the laser intensity). Viewed in a narrow "atomic physics" context the response to the terahertz field might not come as a surprise, but we believe that there is a wider and more important setting. The crucial aspect of our work is that it shows terahertz-frequency fields can induce these phenomena (as noted by Reviewer # 2, "Sharp transitions and bistability are important, regardless of their source/interpretation"), and it is the low intensity required of the terahertz field that makes the work exciting for practical applications. Since preparing the manuscript a proposal highlighting the utility of dissipative quantum systems exhibiting first order phase transitions as sensors has been published [Raghunandan, Wrachtrup, and Weimer, PRL 120, 150501 (2018)]

Manuscript change:

- **Introduction:** New reference [15], Raghunandan 2018.

(2) Do they also observe the critical slowing down phenomena in the terahertz experiment; i.e., when they slightly change the terahertz intensity and observe the recovery, does the recovery time diverge? This experiment may become the final proof that they surely succeeded to observe the terahertz-induced phase transition.

We did indeed observe the transient response becoming slower close to the threshold terahertz intensity, in a way that is qualitatively similar to Carr et al. (Ref. [6]). Our current understanding is that this behaviour is a mean-field-like effect due to approaching the boundaries of the bistability window (the "spinodal lines"), as discussed e.g. in [Marcuzzi, Levi, Diehl, Garrahan, and Lesanovsky, PRL 113, 210401 (2014)].

Manuscript change:

- **Section I, Experiment:** "The response of the system to sudden changes was observed to slow down close to the threshold intensity, in a manner which is qualitatively similar to that previously reported in [6]."

After the above questions are clarified, the paper may be suitable to publish in Nature Communications.

Reviewer #2:

The authors discuss their investigation of optical bistability in a room temperature cell of Rb atoms.

Just to avoid potential misunderstandings, we mention that we used caesium (as stated in the manuscript) rather than rubidium. However, it would be reasonable to assume that analogous results could be achieved with rubidium.

As described in a previous publication [Ref 6] by the same principal investigators, when atoms in the cell are laser excited to the 21P Rydberg state, both the optical transmission and fluorescence from the cell exhibit sharp transitions between low and high values as a function of laser detuning from the Rydberg resonance such that the system exhibits bistability in these characteristic observables vs detuning. The sharp transitions between low and high transmission/fluorescence are attributed to a phase-transition. The addition of a colinear narrow-band THz beam introduces a coupling between the 21P and 21S Rydberg states, thereby modifying the range of detunings at which bistability occurs. Because the degree of modification depends sensitively on the THz field amplitude (and THz frequency), the authors propose to use the effect as a THz detector. Using a mean-field model, the authors attribute the bistable response to the optical and THz field combination as the result of microscopic electric fields, produced by collisional ionization of highly-excited atoms in the vapor, which shift the energies of the Rydberg levels coupled by the optical and THz fields.

This is an accurate summary of the work.

I find both the previously identified bi-stability, and its modification by the THz coupling reported here, to be very interesting effects and the experimental observations are striking. The proposal to use the effect as a THz sensor is, to my knowledge, novel and potentially important.

We thank the reviewer for this positive assessment of the work.

While the manuscript is generally well written, and I do not doubt the correctness of the measurements or their interpretation, I do have a number of comments and questions about the work that I believe are relevant to its ultimate impact in fundamental science and detector development.

1. The title, as well as much of the abstract and introduction, focuses on the THz-driven phase-transition theme, although this notion itself is not critical for the THz-sensor applications that seem to be the primary interest of the authors. Sharp transitions and bistability are important, regardless of their source/interpretation.

The fundamental and applied aspects of the work are both of interest to the authors.

Further, I disagree with the claims in the abstract and introduction: "Here we report a room-temperature phase transition..." As the authors admit later in the introduction, the electronically excited system is non-thermal and, therefore, cannot be characterized by a temperature. The atoms may be at room temperature, but the bound electrons responding to the THz field are certainly not.

We absolutely agree that the electronically excited system is non-thermal and cannot be characterised by a temperature. The point we wish to convey is that our "non-thermal" phase transition is embedded in an otherwise room-temperature system. We thank the Referee for pointing out how our previous wording could have been confusing to the reader.

Comparing the work to studies of phase-transitions (THz driven or otherwise) in thermal systems where the ratio of the interaction energy to the temperature is an important figure of merit, does not seem appropriate.

Thermal, room-temperature systems are very important from a practical point of view because they are the easiest to prepare and have a much reduced experimental overhead when compared to cryogenic or laser-cooling setups. However, we acknowledge that the discussion of equilibrium systems might be confusing to a reader and have modified the manuscript to provide clarification.

Manuscript changes:

- **Title:** “A terahertz-driven **non-equilibrium** phase transition in a room temperature atomic vapour”
- **Abstract:** “Here we report a **non-equilibrium** phase transition driven by a weak ($<1 \text{ Vcm}^{-1}$), continuous-wave terahertz electric field. The system consists of **room-temperature** caesium vapour under continuous optical excitation...”
- **Introduction:** We omit the discussion of equilibrium systems, and now aim to state clearly and unambiguously the non-equilibrium nature of our system.

2. The authors focus on two specific Rydberg states. Have analogous phenomena been observed for other states? If so, why/how were the states which were used chosen? If not, why not? At least a brief discussion of these points would be instructive.

The choice of Rydberg states was informed primarily by the frequency range of the terahertz source. However, similar behaviour was observed for a handful of different Rydberg states within the source range. The detailed study in the manuscript was carried out for only the $21P_{3/2} \rightarrow 21S_{1/2}$ transition but we expect to find qualitatively similar results for other transitions.

Manuscript change:

- **Conclusion:** “While the system would not be suitable for broad-band terahertz sensing, transitions between different Rydberg states span the terahertz-frequency spectrum [20], and a detector could be ‘tuned’ to any of these frequencies by changing the Rydberg state to which the atoms are driven. In general higher energy Rydberg states would require more laser excitation power and correspond to lower terahertz-frequencies, but result in stronger interactions (faster ionisation and stronger polarizability).”

3. The observed effects depend critically on the THz frequency, thus it seems that a THz sensor based on this system would also only operate for a very well-defined THz frequency. Depending on the answer to 2 above, I suspect it would be possible to develop sensors for use at specific, narrow-band discrete frequencies, but not for a continuous band of frequencies or for broader band THz sources. Some discussion of the spectral range and sensitivity of a such a detector should be included.

This is indeed correct. Please see manuscript changes relating to point #2.

4. Given that the bistability is attributed to electric fields due to collisional ionization, some discussion of the influence of external electric fields, including macroscopic fields due to longer lived space charge in the cell, should be included.

The effect of external electric fields and space charge on Rydberg bistability is an interesting and open topic [see eg., Weller et al. PRA 94 063820 (2016) - Ref 8.]. Potentially, the high polarizability of Rydberg atoms might allow for electric fields to tune the sensitivity band of a given Rydberg transition. However, answering these exciting questions is well beyond the scope of the present work.

On a related point, how is the collisional ionization (and therefore the bistability parameters and detector characteristics) affected by other variables such as:

(i) the use of different Rydberg states as the system,

Please see response to point #2 / #3

or (ii) Rb temperature/pressure variations?

The temperature/pressure does change the ionisation rate (see Carr PhD Thesis 2013, Durham University), and therefore affects the phase transition. Please see manuscript changes below.

The authors note that better experimental stability would improve the detector characteristics. Which of the numerous experimental parameters is the biggest limitation on the performance of the two different sensor schemes described: laser frequency/power stability, THz frequency stability, cell temperature, stray electric fields,....?

The main limitation for the latching scheme was the drift of the Rydberg laser frequency which was not stabilised to an atomic reference. This drift was systematically compensated in the second scheme, but still constrained the duration over which data could be collected.

Manuscript change:

- **Section II, Mean-field Model (end):** “We note that the ionisation rate is dependent on vapour pressure and hence the Rydberg phase transition is strongly influenced by the temperature of the vapour.”
- **Section III, Sensing:** “Further work stabilising the Rydberg laser frequency to an atomic reference would allow the vapour to be biased closer to the phase transition, ...”

5. In the mean-field model, why are the spontaneous decay rates Γ_R and Γ_T chosen to be the same? The spontaneous decay rates for Rb S and P Rydberg states are well known and they are quite different from each other. How are the results of the model different if these decay rates are not identical?

There are very many mechanisms depleting the Rydberg population including spontaneous decay, beam crossing, ionisation, black body-induced decay etc. Some of these are well known (spontaneous decay, beam crossing time), but others are much harder to guess (eg. ionisation might be dominant), so it would be inappropriate to claim a specific ratio. The model is intended as for phenomenological intuition rather than a strict simulation, and so the rates are set equal in the interest of restricting the parameter space. Since receiving the manuscript reviews we have repeated the calculations with Γ_R/Γ_T in the range 0.5 \rightarrow 2, and each time reproduced qualitatively similar results. We are therefore confident to claim that the model is robust and does not depend on the fine tuning of parameters.

Manuscript change:

- **Section II, Mean-field Model:** “we phenomenologically describe depletion from the excited state...”
- **Section II, Mean-field Model:** “In the following, we set $\Gamma_R = \Gamma_T = \Gamma$, although the qualitative results of the simulation do not rely on this assumption.”

6. In the mean-field model, how sensitive are the general results to changes in the collisional ionization rates?

The qualitative behaviour is unchanged by small changes to the ionisation rate.

The authors state that sensors based on the bistability phenomenon will not be limited by laser power fluctuations as other techniques are. However, the collisional ionization rates will certainly vary with Rydberg density, which will vary with excitation laser and THz power variations.

The sensitivity of the system to THz power variations is exactly what we are trying to exploit.

Has this effect been considered and analyzed?

We have studied the phase transition with different excitation laser powers and indeed observe corresponding changes to the behaviour (eg. the Rydberg laser power must exceed a certain threshold for the phase transition to take place). An analysis of the implications for detector noise is beyond the scope of this work. However we note that the frequency and intensity of the Rydberg laser were not actively stabilised in our experiment, so we would expect to see less noise (improvement) in the future.

Reviewer #3:

In this manuscript, the authors studied the phase transition of room-temperature atomic cesium vapor switched by weak THz-frequency electromagnetic pulse. They implemented Rydberg atoms in 21P energy level through three-photon process and then coupled to 21S energy level via 0.634 THz field. In this case, the cesium vapor showed a bistability of two steady states (low and high densities in Rydberg energy levels) and the two steady states behaved differently in laser transmission and also in fluorescence, upon being transmitted with the probe laser pulse (one of the three-step excitation lasers to 21P). The authors studied this bistability as a function of detunings of THz field and optical field. The result shows sensitive detection of THz field in V/cm-level of E-field amplitude, sensitive enough for a narrowband terahertz-field detector. The measurements were compared with theoretical calculation using mean-field model taking into account the recently suggested inter-atomic collisions due to ionized Rydberg atoms [Ref. 8], showing an excellent replica qualitatively well agreeing the data.

This is an accurate summary of the work.

The quantum many-body dynamics is an intriguing question in physics which is being received much recent attention, especially in cold atom experiment. The present manuscript deals with the room-temperature bistability behavior of the Rydberg matter and opens the door to application of this fundamental physics to a real sensing application. The present work is in my opinion valuable and timely, and likely to be of interest for a broad audience. Overall the manuscript is well written and understandable, and the method is accessible even for non-specialists. For all these reasons, I recommend publication in Nature Communications.

We thank the reviewer for this positive assessment of our work.

I only have a few minor comments that could help improve the manuscript a bit.

1. Most likely the result is not sensitive to the temperature, but it is worthy at least to mention it in the manuscript.

The temperature determines the number density of the vapour, which in turn influences the ionisation dynamics and the phase transition. However this influence on the phase transition is indirect. (The electronic configuration of the atoms is far from thermal equilibrium and cannot properly be described by a temperature. Instead the number density influences the ionisation rate.) Experimentally we used a servo circuit and heater to stabilise the temperature of the vapour.

Manuscript Change:

- **Section II, Mean-field Model (end):** “We note that the ionisation rate is dependent on vapour pressure and hence the Rydberg phase transition is strongly influenced by the temperature of the vapour.” (see response to reviewer 2)
- **Methods:** “The temperature of the vapour is stabilised around 70C using a servo circuit...”

Probably the Ω^{meas}_T in Page 3 is the “measured” Rabi frequency between $|R\rangle$ and $|T\rangle$, but this is not defined and needs to explain how to measure it.

A description of this measurement is included in the methods – we now direct the reader to the methods at this point in the text.

Manuscript Change:

- **Section I, Experiment:** The Rydberg-transition Rabi frequency is recorded in a separate in-situ measurement of Autler-Townes splitting [26,31] (see methods).

Reviewers' comments:

Reviewer #1 (Remarks to the Author):

The authors clearly answered my questions, and I recommend to publish the article as is in the Nature Communications.

Reviewer #2 (Remarks to the Author):

The authors have revised their manuscript in accord with the questions and comments of the referees, and I believe it has improved as a result. The only previously raised issue that I do not believe has been adequately addressed relates to the detailed influence of various parameters (stray electric fields, temperature, Rydberg density variations due to laser power/frequency variations...) on the proposed THz detector performance. The authors have acknowledged that the system is indeed influenced by these parameters but, perhaps due to the complexity of the processes involved, do not supply an analysis of the corresponding limitations they impose. This is not necessarily unreasonable for this first demonstration except for the fact that they specifically criticize the well-defined limitations of other THz detectors (e.g. probe laser shot noise). In my opinion, they should hold their method to the same level of scrutiny if they make those criticisms/comparisons.

Reviewer #1:

The authors clearly answered my questions, and I recommend to publish the article as is in the Nature Communications.

We thank the reviewer for re-assessing our work in the light of our previous exchange.

Reviewer #2:

The authors have revised their manuscript in accord with the questions and comments of the referees, and I believe it has improved as a result. The only previously raised issue that I do not believe has been adequately addressed relates to the detailed influence of various parameters (stray electric fields, temperature, Rydberg density variations due to laser power/frequency variations...) on the proposed THz detector performance. The authors have acknowledged that the system is indeed influenced by these parameters but, perhaps due to the complexity of the processes involved, do not supply an analysis of the corresponding limitations they impose. This is not necessarily unreasonable for this first demonstration except for the fact that they specifically criticize the well-defined limitations of other THz detectors (e.g. probe laser shot noise). In my opinion, they should hold their method to the same level of scrutiny if they make those criticisms/comparisons.

A detailed analysis (either practical or theoretical) of the noise sources is indeed beyond the scope of this work which is intended as a proof of principle demonstration. Because our work exploits terahertz-driven collective behaviour (which is not only so far unstudied, but also acknowledged to be difficult to model from first principles), we expect the noise limitations to be different from the Rydberg electrometry technique that we refer to in the manuscript [26,28,38-40]. For example, we would reasonably expect the laser shot noise (which limits Rydberg electrometry) not to place such a strict limit on our system (due to the higher laser intensity). However, we acknowledge that it would be wrong to imply that there are definitely no other equivalent limitations of our system which are as yet undiscovered.

It is our intention to state clearly the achievements and limitations of the previous work, and report the noise performance that we have seen in our detector so far. However, we try to avoid drawing comparisons using anything of our own work that we have not explicitly demonstrated in the manuscript. For this reason, we can now see that our previous choice of wording at the beginning of Section III may have been misleading, and thank the Referee for pointing this out to us. We have revised the paragraph, and omit any direct comparison of the signal to noise ratio with reference to shot noise. To the best of our knowledge we have given a fair account of shortcomings, imperfections and error sources of our method and we hope that with the rectification of the wording the manuscript is now publishable.

Manuscript change (Section III): *“Room-temperature atomic vapour holds particular promise as a medium for measuring terahertz fields. The well-known atomic properties allow for absolute calibration to SI units [26], and rapid recent progress in Rydberg electrometry [28, 28, 39] has delivered unprecedented sensitivity in the microwave [26] and mm-waves range [40]. This technique has been shown to be only limited by the shot noise of the probe laser [39], the intensity of which needs to be kept low to avoid spectral broadening effects. Here we report proof of principle terahertz field measurements that harness the collective behaviour of the vapour. Near the phase transition the abrupt changes in the optical response of the medium provide sharp spectral features even at high probe laser intensity, yielding a fast and sensitive method for measuring narrowband terahertz radiation.”*

REVIEWERS' COMMENTS:

Reviewer #2 (Remarks to the Author):

The authors have responded to my concerns and modified their manuscript accordingly. I recommend publication of the revised manuscript in Nature Communications.

Reviewer #2:

The authors have responded to my concerns and modified their manuscript accordingly. I recommend publication of the revised manuscript in Nature Communications.

We thank the reviewer for re-assessing our work in the light of our previous exchange.